# Positioning of Minimally Invasive Liver Surgery for Hepatocellular Carcinoma: From Laparoscopic to Robot-Assisted Liver Resection

**DOI:** 10.3390/cancers15020488

**Published:** 2023-01-12

**Authors:** Shogo Tanaka, Shoji Kubo, Takeaki Ishizawa

**Affiliations:** Department of Hepato-Biliary-Pancreatic Surgery, Osaka Metropolitan University Graduate School of Medicine, Osaka 545-8585, Japan

**Keywords:** hepatocellular carcinoma, laparoscopic liver resection, long-term outcomes, robot-assisted

## Abstract

**Simple Summary:**

Laparoscopic liver resection is widely accepted in the surgical treatment of hepatocellular carcinoma. Laparoscopic liver resection has been reported to result in earlier postoperative recovery and fewer postoperative complications than open liver resection for hepatocellular carcinoma. Laparoscopic liver resection is technically feasible for selected patients with hepatocellular carcinoma even under several situations such as the prevalence of liver cirrhosis, obesity, elderly, hepatocellular carcinoma recurrence (repeat liver resection), and major resection that led to better intra- and post-operative outcomes than open liver resection. In recent years, robot-assisted liver resection has gradually become popular, and its short- and long-term results for hepatocellular carcinoma are reported to be not different from those of laparoscopic liver resection. Robot-assisted liver resection is expected to become the mainstay of minimally invasive surgery in the future.

**Abstract:**

Laparoscopic liver resection (LLR) is widely accepted in the surgical treatment of hepatocellular carcinoma (HCC) through international consensus conferences and the development of difficulty classifications. LLR has been reported to result in earlier postoperative recovery and fewer postoperative complications than open liver resection (OLR) for HCC. However, the prevalence of liver cirrhosis, obesity, the elderly, HCC recurrence (repeat liver resection), and major resection must be considered for LLR for HCC. Some systematic reviews, meta-analysis studies, and large cohort studies indicated that LLR is technically feasible for selected patients with HCC with these factors that led to less intraoperative blood loss, fewer transfusions and postoperative complication incidences, and shorter hospital stays than OLR. Furthermore, some reported LLR prevents postoperative loss of independence. No difference was reported in long-term outcomes among patients with HCC who underwent LLR and OLR; however, some recent reports indicated better long-term outcomes with LLR. In recent years, robot-assisted liver resection (RALR) has gradually become popular, and its short- and long-term results for HCC are not different from those of LLR. Additionally, RALR is expected to become the mainstay of minimally invasive surgery in the future.

## 1. Introduction

Hepatocellular carcinoma (HCC) is the most common primary liver tumor and the third leading cause of cancer-related death worldwide [1,2]. Liver resection is a valuable treatment modality in patients with HCC with preserved liver function [3,4]. The first laparoscopic liver resection (LLR) was reported in 1992, whereas the first LLR for HCC was in 1995 [5]. The LLR application was considered controversial for many years. Progress in laparoscopic techniques and expertise in combination with technological advances have led to more widespread adoption of minimally invasive approaches for HCC resection over the last 15 years [6]. Subsequently, the number of LLR cases increased due to the roadmap advocacy for the widespread use of safe LLR at numerous international consensus conferences [7,8,9,10] and the development of a difficulty scale classification [11,12,13,14]. Additionally, in Japan, the number and the proportion of LLR for the total number of liver resections increased from 1848 cases (9.9%) in 2011 to 5648 (24.8%) in 2017 [15]. At present, solitary lesions (≤5 cm) located in segments 2 through 6, which was the most acceptable LLR indication, as well as laparoscopic major liver resection, have been performed [7,8,14,16,17,18,19,20]. With these LLR developments, perioperative outcomes are better in patients with HCC who underwent LLR than those who underwent OLR, with no difference in long-term outcomes [16,17,21,22,23], whereas a recent systematic review and meta-analysis study indicated better long-term outcomes after minimal invasive liver resection (MILR), including LLR (48 articles) and robot-assisted liver resection (RALR, 2 articles) for HCC than OLR among the recently published data [24]. The pooled analysis revealed an 18% decrease in disease-specific 3-year mortality after MILR (almost, LLR) compared with OLR (Figure 1), and the sensitivity analysis of contemporary studies from 2010 to 2019 revealed a significantly lower 5-year all-cause mortality and 3-year disease-specific mortality in MILR compared to OLR. Thus, the overall picture is important in the surgical HCC treatment; however, factors such as cirrhosis due to background liver disease, repeat liver resection for HCC recurrence, advanced age, and obesity must be considered.

We reviewed the short- and long-term results of LLR usefulness (vs. OLR) with a special focus on these factors. Additionally, the usefulness of RALR, which has become increasingly popular in recent years, is discussed.

## 2. Liver Cirrhosis

Most patients with HCC commonly have chronic hepatitis and cirrhosis. Liver resection for patients with cirrhosis is challenging due to elevated portal pressure and impaired coagulation function. One systematic review and meta-analysis [59], one systematic review [60], and two meta-analyses [61,62] compared LLR with OLR for patients with cirrhosis with HCC. These reports revealed no difference in operation time among patients who underwent LLR and OLR; however, LLR reports decreased blood loss, transfusion rate, postoperative complications (including postoperative ascites and liver failure), and length of hospital stay. Moreover, LLR gains better 1-year overall survival (OS) [61,62] and 5-year OS [60,61,62]. Only one report revealed better 1-year disease-free survival (DFS) in LLR than in OLR [61]. However, among patients with cirrhosis, patients with Child-Pugh class B were reported to have more complications and deaths in the hospital and poorer long-term outcomes than patients with Child-Pugh class A [63,64,65,66,67], but the effect of LLR remains controversial because of the small number of patients [68,69]. Recently, Berardi et al. [70] reported an international multicenter study of 253 patients with Child-Pugh class B regarding short- and long-term outcomes. The comorbidity prevalence, increased Child-Pugh score (7 to 9), decreased preoperative hemoglobin and platelet count, and preoperative ascites and portal hypertension prevalence, increased the risk for postoperative complication within 90 days postoperatively (Figure 2). Moreover, minimally invasive surgery, including LLR and minor liver resection, decreased the risk for postoperative complications. Additionally, LLR did not affect DFS or OS rates. Liver cirrhosis is a well-known risk factor for postoperative liver failure-related mortality [71]. However, the development of devices, hemostasis techniques, and pneumoperitoneum and minimization of delamination in the LLR has controlled the bleeding and prevented postoperative ascites [25,72], which might lead to postoperative early recovery even for patients with Child-Pugh class B cirrhosis. Some better LLR prognoses might be caused by less compression during laparoscopic manipulation, which prevented tumor cell metastasis [62]. However, several reports revealed that LLR has no effect on long-term prognosis (no difference from OLR) [25,73], and only tumor factors were found to determine DFS in a study of Child-Pugh class B, while tumor factors and systemic status, including cirrhosis, determine OS [70]. LLR may be a useful treatment for patients who may not have previously been candidates for open surgery and may even prolong survival. However, further study is needed on the efficacy of LLR on long-term outcomes after cirrhotic liver resection.

## 3. Laparoscopic Repeat Liver Resection (LRLR) for Recurrent HCC

High recurrence even after curative liver resection for the initial HCC is a significant oncologic feature of HCC [74,75,76,77]. Additionally, hepatic resection is recommended for HCC recurrences (HCCR), as well as primary cases, if HCC has ≤3 nodules [4]. However, adhesions after initial hepatectomy are not only seen on the liver dissection surface, but also on the dissection surface and hepatoduodenal mesentery at a certain frequency, which makes repeat liver resection difficult, leading to unexpected blood loss and vascular or biliary structure intraoperative injury [78,79,80]. Conversely, some reported the remits of LRLR, such as minimalization of dissection of the adhesion under high magnification directly from the caudal direction [13,81] and small targeted area without damages to the surrounding area in the LRLR [79]. Some highly experienced centers reported feasible and safe LRLR for HCCR in the single-arm study [81,82,83,84]. A meta-analysis revealed that LRLR (*n* = 145) had a lower rate of in-hospital complication, much less blood loss, and a shorter hospital stay than open repeat liver resection (ORLR, *n* = 190) [85]. However, these studies were very small in number. Recently, an international collaborative study by Morise et al. [86] examined the usefulness of LRLR (*n* = 648) for HCCR and compared ORLR (*n* = 934) using propensity score matching (PSM, each, *n* = 238). The operation time was longer in the PSM cohort (mean, 273 min vs. 232 min, *p* = 0.007), but blood loss was lower (mean, 268 mL vs. 497 mL, *p* = 0.001) in patients who underwent LRLR than in those who did ORLR. No differences were found in the incidence of postoperative 90-day complications, 90-day mortality, length of hospital stay, or long-term survival. Therefore, case selection that would benefit from LRLR would be important. Kinoshita, et al. [87] reported the difficulty of LRLR in 60 patients with HCCR. Additionally, (1) an open approach during previous liver resection, (2) two or more previous liver resections, (3) a history of previous liver resection with not less than a sectionectomy, (4) a tumor near the resected site of the previous liver resection, and (5) intermediate or high difficulty in the difficulty scoring system [11] were independent risk factors for prolonged operative time and/or severe adhesion of LRLR. Thereafter, they validated less blood loss and lower postoperative complication incidence in LRLR than in ORLR among patients with ≤3 applicable risk factors; however, the operation time was longer in LRLR than in ORLR, and no difference was observed in other intra- and postoperative outcomes among LRLR and ORLR in patients with ≥4 of these 5 variables, suggesting that LRLR has no advantage in these patients [88]. On the basis of these findings, LRLR may have better short-term results than ORLR, but preoperative evaluation, such as details of prior surgeries, will be needed to determine whether it can be safely applied.

## 4. Elderly

The geriatric population has dramatically increased, and the number of elderly patients who undergo liver resection has even more rapidly increased [89]. Some reports revealed that the incidences of postoperative complication and mortality were comparable between elderly and non-elderly patients in OLR [90,91], but others have revealed an increased mortality incidence in elderly patients [92]. The reported incidence of overall postoperative complications in the elderly (aged 65–75 years) ranged from 29% to 59%, that of major complications (Clavien–Dindo grade ≥ IIIa) ranged from 16% to 41%, and that of mortality ranged from 0% to 9% [90,91,92,93]. Large-scale data from the Diagnosis Procedure Combination database, a national administrative database in Japan (2007–2012, *n* = 27,094), indicated the incidence of postoperative complication and mortality after liver resection increased up until the 70 s; however, no differences were found among patients aged in their 70 s, 80–84 years, and ≥85 years [94]. These results may be attributed to the fact that the adaptation is strictly handled for the elderly. Nomi et al. [95] reported a lower incidence of overall postoperative and major complications in elderly patients (aged ≥75 years) with HCC who underwent LLR than in those who underwent OLR, but others reported no difference among LLR and OLR [96,97]. However, LLR shortened the length of hospital stay [95,96,97]. One systematic review and meta-analysis using 12 studies (LLR: *n* = 831 and OLR; *n* = 931) indicated that LLR decreased the intraoperative blood loss, incidence of overall postoperative complications, including liver failure, ascites, and surgical site infection, major complication, and length of hospital stay although it includes all diseases, not just HCC [98]. Therefore, age would not be a determining factor for surgery. However, the high incidence of “elderly-related events”, including respiratory complications (pneumonia and respiratory failure requiring reintubation) [91], cardiac events [90], delirium [90,99], and discharge to rehabilitation facilities [99] are a major problem for liver resection in the elderly. LLR was reported to decrease the incidence of elderly-related events such as cardiopulmonary complications [95,96,100]. Moreover, maintenance of independence after liver resection is very important for elderly patients who underwent liver resection. Our previous study indicated that LLR decreased the incidence of postoperative loss of independence during the early postoperative period, including transfer to rehabilitation facilities, readmission within 30 days, discharge with any health care supports, and/or death within 90 days except cancer-related death, and at 1 year after liver resection, including the need of any healthcare supports and/or death due to deterioration of physical function [101,102]. A few studies reported regarding long-term survival; however, no differences were found in DFS or OS rate among elderly patients with HCC who underwent LLR or OLR [97]. LLR for the elderly has better intraoperative outcomes and fewer postoperative complications than OLR. In addition, LLR may have advantages to reduce elderly-related events and maintain independent living.

## 5. Obesity

The prevalence of obesity and its associated diseases has remained increasing worldwide. The prevalence of obesity (body mass index [BMI] of ≥30 kg/m^2^) is 40% in the United States [103] and approximately 20% in Europe [104]. In Japan, obesity is defined by a BMI of ≥25 kg/m^2^ [105]. As of 2018, 32.2% of males and 21.9% of females aged ≥20 years were classified as obese [106]. Furthermore, several reports revealed that patients with obesity are at high risk of developing HCC [107,108]. Thus, a higher prevalence of obesity and expansion of liver resection indications could increase the number of liver resections among patients with obesity with HCC in the future. Obesity is correlated with comorbidities and technical difficulties in open surgery and is considered a risk factor for postoperative complications in several surgical fields [109,110]. Countermeasures for the depth of the surgical field and large volume of intraperitoneal fat are important in abdominal surgery, including liver resection, in patients who are overweight and obese [111,112]. These situations are associated with increased operation time, blood loss, and postoperative complications in the OLR [113,114,115]. Liver parenchyma dissection and hepatic hilum treatment are sometimes challenging despite a large skin incision and gastrointestinal tract and greater momentum compression in OLR [116,117,118]. In contrast, pneumoperitoneum, head-up position, and high magnification—even at deep portions in the caudal view—can provide sufficient free space to control the forceps in LLR, even in patients who are overweight and obese (Caudal approach, Figure 3) [119,120,121]. There is some disagreement as to whether obesity increases the risk of conversion [12,111,113,122,123], but the LLR is reported to decrease intraoperative blood loss and postoperative complications compared with OLR even in obesity [113,118,121,124]. Moreover, obesity did not affect conversion rate, operation time, or blood loss in the LLR compared with non-obesity [113,122,123]. There is some disagreement regarding conversion to open surgery, but LLR has better short-term outcomes than OLR. Therefore, LLR for obesity would be feasible and safe.

## 6. Robot-Assisted Liver Resection (RALR)

RALRs are slowly spreading, although at a slower speed than LLRs [125,126]. In 2018, an international expert panel published a consensus guideline on the use of robotics in liver surgery, concluding that “RALR is as safe and feasible as LLR and OLR” for both major and minor liver resection [127]. Advantages of RALR include stability and magnification of a three-dimensional view, the best possible ergonomics, enhanced suturing capacity, the ability to complete more extensive or complex minimally invasive operations, integrated fluorescence guidance, and a shortened learning curve. However, the robotic platform remains limited by a paucity of parenchymal transection devices, a complete lack of hepatic feedback, and an additional operation time associated with docking and instrument exchange [128,129].

Some reported learning curves for LLR in 35 to 75 cases regarding operation time and incidence of liver injury (liver ischemia, congestion, or portal vein thrombosis) [130,131,132,133]. Conversely, early proponents of the robotic platform felt that robotic operations would be easier to learn than their laparoscopic counterparts due to the intentionally intuitive nature of robotic instrument us even for novice surgeons [134,135,136]. Some studies indicated shortened learning curves of 15 to 52 cases in RALR [137,138,139]. Additionally, the best possible ergonomics would increase the number of major hepatectomies and/or highly difficult cases [140,141,142]. However, RALR may become mainstream in the future. Some meta-analyses indicated less blood loss and a lower proportion of transfusion and incidence of postoperative complications in patients who underwent RALR than OLR [143,144,145]. Moreover, Kamarajah et al. [146] reported a systematic review and meta-analysis that included 26 articles and 2630 patients (RARL: 950 patients and LLR: 1680 patients) and revealed that blood loss was less (median, 286 mL vs. 301 mL, *p* < 0.001) and operation time was longer (median, 281 min vs. 221 min, *p* < 0.001) in patients who underwent RALR than in those who underwent LLR. Additionally, no difference was found in the incidence of postoperative complications, mortality, or length of hospital stay among patients who underwent RALR and LLR although readmission was lower in patients who underwent RALR than in those who underwent LLR. Moreover, a meta-analysis for major hepatectomy revealed an association between RALR and lower blood loss and conversion rate but with a slightly longer hospital stay compared to LLR [147]. Zhu et al. [148] revealed intra- and postoperative outcomes among patients who underwent RALR (*n* = 71), LLR (*n* = 141), and OLR (*n* = 157) for HCC; operation time was shortest and the length of hospital stay was longest in patients who underwent OLR, and similar results were demonstrated between those who did RALR and LLR. Conversely, some studies reported a higher incidence of postoperative bile leakage after RALR [149,150,151]. RALR is easy to manipulate in the hepatic hilum, but the lack of tactile sensation may cause inadvertent bile duct injury. In contrast, careful infraphrenic dissection was reported to reduce the incidence of postoperative pleural effusions [150]. Therefore, RALR does not significantly differ from LLR and is considered less invasive than OLR in terms of short-term results. Few studies reported on long-term outcomes after RALR; however, Zhu et al. [148] revealed no difference in DFS or OS among patients who underwent RALR, LLR, and OLR. Hence, RALR is as good as LLR as MIS. RALR may provide better perioperative results than LLR with further equipment development.

## 7. Conclusions

In conclusion, liver resection for HCC requires consideration of various situations, such as liver cirrhosis, repeat liver resection, obesity, and the elderly, but LLR overcomes these situations and has equal or better outcomes compared to OLR. In the future, RALR is expected to develop as an MIS alongside LLR.

## Figures and Tables

**Figure 1 cancers-15-00488-f001:**
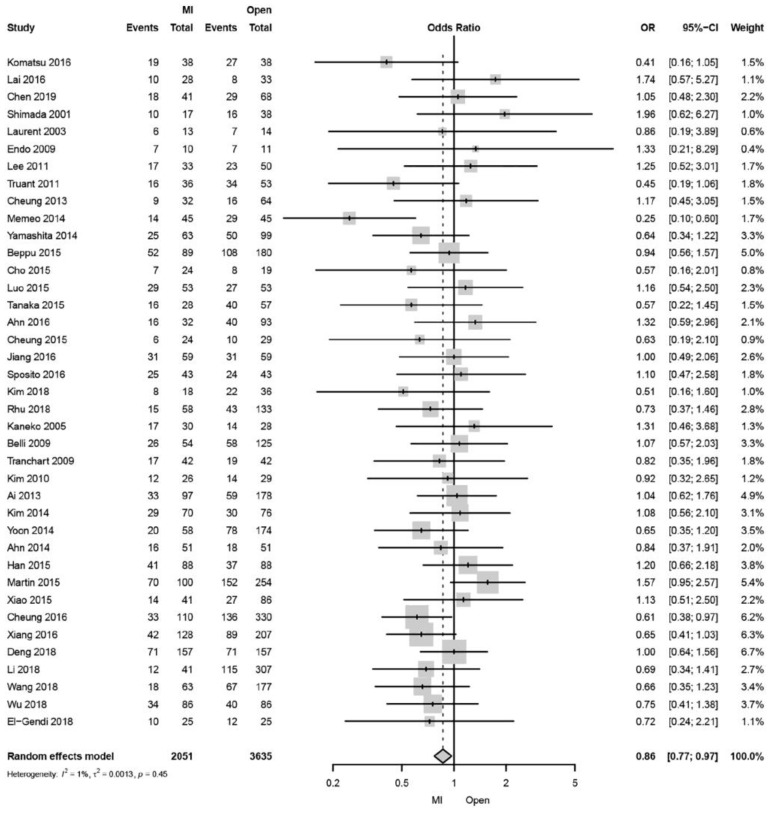
Forest plot of disease-specific 3-year mortality comparing minimally invasive and open liver resection for hepatocellular carcinoma. The studies shown in this figure can be found as references [6,21,23,25,26,27,28,29,30,31,32,33,34,35,36,37,38,39,40,41,42,43,44,45,46,47,48,49,50,51,52,53,54,55,56,57,58]. Reprinted/adapted with permission from Ref. [24]. 2021, SAGE Publications.

**Figure 2 cancers-15-00488-f002:**
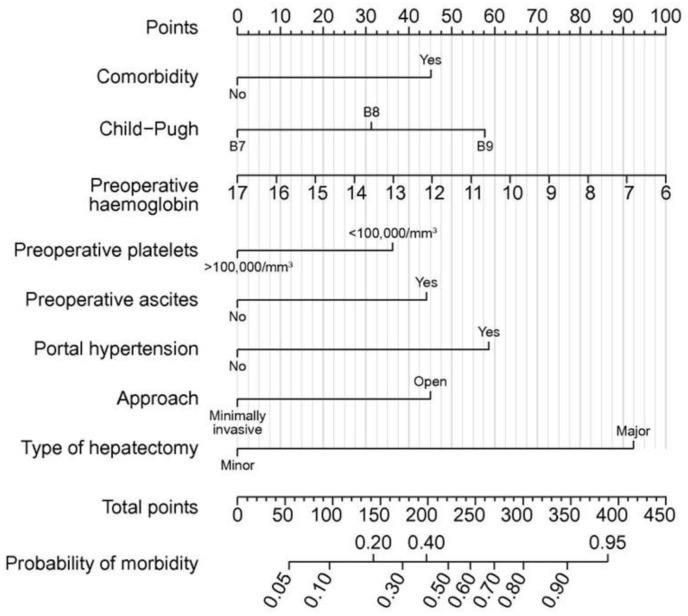
Nomogram for predicting 90-day morbidity after liver resection for hepatocellular carcinoma in patients with Child-Pugh class B. Nomogram was drawn using the multivariable logistic model for 90-day morbidity. Reprinted/adapted with permission from Ref. [70]. 2019, Elsevier.

**Figure 3 cancers-15-00488-f003:**
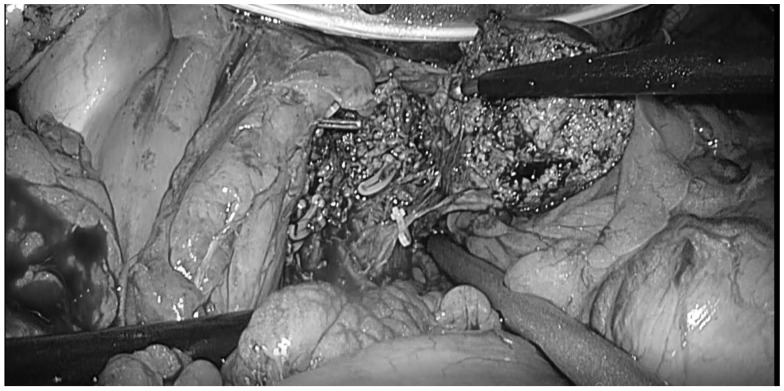
Laparoscopic liver resection for tumor located at segment I in a patient with obesity. Taking advantage of the Caudal approach of laparoscopic surgery, liver resection was performed with a good field of view despite the surgical depth.

## Data Availability

Data sharing is not applicable to this article as no datasets were generated or analyzed during the current study.

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
