# Peer review of "Positioning of Minimally Invasive Liver Surgery for Hepatocellular Carcinoma: From Laparoscopic to Robot-Assisted Liver Resection"

_cancers, 2023, doi:10.3390/cancers15020488_

Round 1

Reviewer 1 Report

Thank you so much for the opportunity to review the paper.

It is an interestig up-to date overview  about the minimally invasive liver surgery for hepatocellular cancer (HCC), however the paper isn't a review, but only an Editorial or an Up-to-Date Overview about the use of minimally invasive liver surgery for HCC.

For a review it is necessary: PRISMA CHECKLIST AND FLOWCHART and also PROSPERO registration. There aren't any data analysis. The authors have focused on some specific issues: liver cirrhosis, laparoscopic repeat resection for recurrent HCC, elderly, obesity, robot assisted liver resection.   - The authors listed only many scientific papers for each section, but what are each chapter's conclusions?   - Where is the Discussion?   The authors do not show a clear position about the topics discussed!!!!

Author Response

Response to Reviewer 1

Thank you very much for considerable comments.

Comment 1: It is an interesting up-to date overview about the minimally invasive liver surgery for hepatocellular cancer (HCC), however the paper isn't a review, but only an Editorial or an Up-to-Date Overview about the use of minimally invasive liver surgery for HCC. For a review it is necessary: PRISMA CHECKLIST AND FLOWCHART and also PROSPERO registration. There aren't any data analysis. The authors have focused on some specific issues: liver cirrhosis, laparoscopic repeat resection for recurrent HCC, elderly, obesity, robot assisted liver resection.   - The authors listed only many scientific papers for each section, but what are each chapter's conclusions?   - Where is the Discussion?   The authors do not show a clear position about the topics discussed!!!!

Response: As you indicated, this paper is not a Systematic Review, but rather an Editorial or Up-to-Date Overview. Therefore, the requirements for PRISMA CHECKLIST AND FLOWCHART are not met. We added the discussion and conclusion in each chapter.

Therefore, we changed the sentence in each chapter

  • Chapter 2: (lines 99 to 102):

“Further study is needed on the efficacy of LLR on long-term outcomes after cirrhotic liver resection.”

To

“LLR may be a useful treatment for patients who may not have previously been candidates for open surgery, and may even prolong survival. However, further study is needed on the efficacy of LLR on long-term outcomes after cirrhotic liver resection.”

  • Chapter 3: (lines 138 to 141)

“Therefore, proceeding with caution will be necessary for LRLR while ascertaining the details of prior surgeries.”

To

“On basis of these findings, LRLR may have better short-term results than ORLR, but preoperative evaluation, such as details of prior surgeries, will be needed to determine whether it can be safely applied.”

  • Chapter 4: (lines 178 to 180)

“LLR for elderly patients with HCC would have merit for suppressing age-related events and loss of independence.”

To

“LLR for the elderly has better intraoperative outcomes and fewer postoperative complications than OLR. In addition, LLR may have advantages to reduce elderly-related events and maintain independent living.”

  • Chapter 5

We added the sentence on lines 205 to 206:

“There is some disagreement regarding conversion to open surgery, but LLR has better short-term outcomes than OLR.”

  • Chapter 6.

We already described the discussion and conclusion on lines 253 to 255.

Reviewer 2 Report

Congratulations for the authors in writing this paper in an interesting area of liver resections for HCC. The work is well organized and well described.The message is clear and the work design is appropriate to answer the research question

Author Response

Response to Reviewer 2

Comment 1: Congratulations for the authors in writing this paper in an interesting area of liver resections for HCC. The work is well organized and well described. The message is clear and the work design is appropriate to answer the research question.

Response: Thank you very much for your kind comment.
